# BERTAQA: How Much Do Language Models Know About Local Culture?

**Julen Etxaniz**    **Gorka Azkune**    **Aitor Soroa**    **Oier Lopez de Lacalle**    **Mikel Artetxe**
HiTZ Center – Ixa, University of the Basque Country (UPV/EHU)
`julen.etxaniz@ehu.eus`

## Abstract

Large Language Models (LLMs) exhibit extensive knowledge about the world, but most evaluations have been limited to global or anglocentric subjects. This raises the question of how well these models perform on topics relevant to other cultures, whose presence on the web is not that prominent. To address this gap, we introduce BERTAQA, a multiple-choice trivia dataset that is parallel in English and Basque. The dataset consists of a local subset with questions pertinent to the Basque culture, and a global subset with questions of broader interest. We find that state-of-the-art LLMs struggle with local cultural knowledge, even as they excel on global topics. However, we show that continued pre-training in Basque significantly improves the models' performance on Basque culture, even when queried in English. To our knowledge, this is the first solid evidence of knowledge transfer from a low-resource to a high-resource language. Our analysis sheds light on the complex interplay between language and knowledge, and reveals that some prior findings do not fully hold when reassessed on local topics. Our dataset and evaluation code are available under open licenses at `https://github.com/juletx/BertaQA`.

## 1  Introduction

Large Language Models (LLMs) have obtained impressive results on a wide range of tasks, with many benchmarks being solved soon after being released [Team et al., 2023, OpenAI et al., 2024]. Nevertheless, the majority of language model research is conducted in English, and the evaluation of these models has predominantly focused on anglocentric or global subjects. For instance, GPT-4 was reported to obtain human-level performance on a wide range of professional and academic exams [OpenAI et al., 2024], but the majority of these exams belong to US programs.[1] Furthermore, multilingual benchmarks tend to suffer from the same issue, as most of them are created by translating English datasets into other languages [Conneau et al., 2018, Artetxe et al., 2019, Bandarkar et al., 2023]. As such, the current evaluation of LLMs barely covers topics that are idiosyncratic to other cultures, falling short at measuring the true usefulness of LLMs for users from these communities.

To better assess how LLMs perform on local topics from a minority culture in comparison with global topics, we introduce BERTAQA.[2] BERTAQA is a multiple-choice trivia dataset with 4,756 questions divided into two subsets: local questions about the Basque Country and its culture,[3] and global questions about subjects of broader interest. These questions were originally authored in Basque and professionally translated into English, making the dataset fully parallel in these two languages. The questions cover 8 diverse categories, and are labeled as easy, medium or hard. As shown in Table

---

[1]In particular, 33 out of 34 exams correspond to programs or organizations from the US or Canada, such as UBE, GRE or AP, and the remaining one corresponds to coding exercises.

[2]*BertaQA* is pronounced similarly to the Basque word *bertakoa*, which means *local*.

[3]Located on the western edge of the Pyrenees, straddling northern Spain and southwestern France, the Basque Country is a region with a distinctive culture and language—Basque or Euskara, a low-resource language isolate.

Table 1: Examples from the English version of BERTAQA for each subset and category.

| | Local Questions | Global Questions |
|---|---|---|
| **Basque and Literature** | What does the "Karmel" magazine specialize in?
a) Bertsolarism
**b) Basque culture in the past and the present**
c) The life of the Carmelites | In which of these novels does the sea not appear?
**a) "The Adventures of Tom Sawyer"**
b) "Moby Dick"
c) "Treasure Island" |
| **Geography and History** | Where's Atxondo?
**a) In Biscay**
b) In Gipuzkoa
c) In Navarre | Who was imprisoned in 1964?
**a) Nelson Mandela**
b) Mumia Abu Jamal
c) Charles Ghankay |
| **Society and Tradition** | Which of the following is a Basque Government institution?
a) IKA
b) AEK
**c) HABE** | What kind of energy do we use most?
**a) Oil**
b) Hydroelectric power
c) Nuclear power |
| **Sports and Leisure** | Where was Julian Retegi born?
a) Areso
**b) Eratsun**
c) Eraso | Which country has won the most FIFA World Cup titles?
a) Argentina
b) Germany
**c) Brazil** |
| **Culture and Art** | Who built the Gaztelu Berria or Château-Neuf in Bayonne?
**a) The English**
b) The French
c) The Spanish | When did the Titanic Belfast Museum open?
**a) In 2012**
b) In 2005
c) In 2002 |
| **Music and Dance** | Where did the dance called "Dantzari" originate?
a) In the Busturia area
b) In the Enkarterri area
**c) In the Durango area** | Who wrote the soundtrack for the James Bond series?
**a) John Barry**
b) Henry Mancini
c) John Williams |
| **Science and Technology** | Which town in Biscay is associated with dynamite?
a) Leioa
**b) Galdakao**
c) Erandio | What is the scientific name for daltonism?
a) Chondrostoma
b) Chromatosis
**c) Dyschromatopsia** |
| **Cinema and Shows** | What's the name of the film based on Bernardo Atxaga's novel "Obabakoak"?
a) "Obabakoak"
**b) "Obaba"**
c) "Obabako istorioak" | What instrument did Marilyn Monroe play in the film "Some like it hot"?
a) The Harp
b) The Didgeridoo
**c) The Ukulele** |

1, the local subset includes questions like the birthplace of Julian Retegi (a renowned champion of *Basque pelota*, a local sport), while the global subset covers topics like the soundtrack of James Bond.

Our experiments show that existing LLMs perform much better on global topics than on local topics. For instance, GPT-4 Turbo obtains 91.7% accuracy on the global subset and 72.2% on the local subset. In addition, we find that continued pretraining in Basque can substantially improve the performance on the local subset at the cost of some degradation on the global subset. For example, we outperform Llama 2 70B by 13.5 points on the local subset by continuing training it on Basque data, while losing 4.1 points on the global subset. This shows that evaluating on global questions alone, as it is commonly done, can show a distorted picture, as the trends can be radically different on local questions. Similarly, we find that translation-based approaches like *translate-test* [Conneau et al., 2018] and *self-translate* [Etxaniz et al., 2023] are much more effective on global questions. All in all, our results prompt to reconsider some prior findings when reevaluated on local subjects, and demonstrate the complex interplay between language, knowledge and culture.

In summary, our paper makes the following contributions:

- We release BERTAQA, a multiple-choice trivia dataset with 4,756 questions divided into two subsets: a local subset with questions pertinent to the Basque culture, and a global subset with questions of broader interest.
- We evaluate a wide range of open and commercial models and show their limitations on local questions, where they obtain significantly worse results.
- We show that continued pretraining in Basque substantially improves the models' knowledge of the Basque culture, even if queried in English. This proves that it is possible to transfer knowledge from a low-resource to a high-resource language.
- We show that LLMs fail to encode knowledge in a fully language-agnostic manner, and perform better when queried in the language they acquired the relevant knowledge in—favoring Basque for local questions and English for global questions.

- We show that translate-test and self-translate work better for global questions than local questions, demonstrating that these approaches are not always as effective as reported in prior work.

## 2 BERTAQA

BERTAQA is a trivia dataset comprising 4,756 multiple-choice questions, with a single correct answer and 2 additional distractors. Crucially, questions are distributed between *local* and *global* topics. Local questions require specific knowledge about the Basque Country and its culture, while global questions require more general world knowledge. Additionally, questions are classified into eight categories: Basque and Literature, Geography and History, Society and Traditions, Sports and Leisure, Culture and Art, Music and Dance, Science and Technology, and Cinema and Shows. Questions are also labeled according to their difficulty as easy, medium or hard. Table 1 shows examples of BERTAQA.

The dataset was originally compiled in Basque by crawling public sources that are no longer available. The questions were already classified into local and global topics, and labeled according to their difficulty level and knowledge category. We inspected the dataset and confirmed that the division was well-founded. The motivation for using this global subset, as opposed to existing QA datasets, is that the questions come from the same source, so the results should be more comparable. As such, no human annotator was involved in the creation of the Basque portion of the dataset. To check whether the content is present in other websites, we wrote some of the questions verbatim in Google search using quotation marks, but received no results. Finding the answer to some of these questions on the web is still possible, but at least the same questions are not present on the web. We do this experiment in Section 4.5, where we try to find the correct answer of some questions by searching the web. While this cannot categorically discard contamination, we believe that this, along with the nature of the raw data we crawled and the results from our experiments, makes it very unlikely that existing models were exposed to the same data during training.

Starting from the original version in Basque, we also created an English version of BERTAQA using a professional translation service (Elhuyar itzulpenak[4]). We first wrote some translation guidelines, covering things like formatting or using Wikipedia as a reference when available to translate Basque named entities. In addition, our guidelines asked translators to discard questions whose answers require knowing the Basque language (such as onomatopeias). We initially sent 100 question/answers for translation. We reviewed these translations carefully, and worked closely with the professional translators to clarify and extend the guidelines accordingly. The remaining dataset was translated in batches of 1000 samples. The translators tagged problematic samples (difficult translation, outdated information, more than one correct answer...), which we manually reviewed. During the translation process, a few of the original questions in Basque were corrected, either because the original answer was incorrect or it became outdated. In addition, we discarded a few questions that required knowledge of Basque or English, and would lose their essence if translated.

The resulting dataset is balanced regarding the number of questions per category and subset, with around 300 questions in each. The number of questions per difficulty is also balanced: most categories have around 110 easy and medium questions and 80 difficult questions in each subset. The average length of the questions and the candidates is around 50 and 13 characters, respectively. The detailed statistics of the dataset are reported in Appendix A.

## 3 Experimental Settings

We evaluate a wide range of open and commercial models on the BERTAQA dataset, and measure the behavior of those models when answering local and global questions. We start by describing the tested models (Section 3.1), followed by the methods used in our experiments for each model type (Section 3.2).

---

[4] https://itzulpenak.elhuyar.eus/en

### 3.1 Models

We experiment with a wide range of recent models, including open base models and commercial chat models. The models include:

**Open Models.** We tested the recent strongest base models that are publicly available, which are primarily trained in English, but show some multilingual capabilities too: Llama 2 [Touvron et al., 2023], Llama 3 [Meta, 2024], Mistral-7B [Jiang et al., 2023], Mixtral-8x7B [Jiang et al., 2024], Yi [01.AI et al., 2024], Qwen 1.5 [Bai et al., 2023] and Gemma [Team and Deepmind, 2024]. We decided to leave multilingual models like XGLM [Lin et al., 2022] and BLOOM [Scao et al., 2023] out, as they performed close to random performance in our preliminary experiments.

**Commercial Models.** We focus on the leading models from OpenAI and Anthropic. Unlike open models, these models are chat models and include more languages. For OpenAI models, we tested the latest GPT3.5 Turbo (`gpt-3.5-turbo-0125`), GPT4 Turbo (`gpt-4-0125-preview`) and GPT4 (`gpt-4-0614`) [OpenAI et al., 2024]. For Anthropic models, we also tested the most recent models: Claude 3 Opus (`claude-3-opus-20240229`), Claude 3 Sonnet (`claude-3-sonnet-20240229`), Claude 3 Haiku (`claude-3-haiku-20240307`).

### 3.2 Methods

**Open Models.** We evaluated open models using the LM Evaluation Harness library [Gao et al., 2023]. We used the same multiple-choice prompt as Etxaniz et al. [2024] for Basque, and a translated version of it for English (see Appendix C). Following common practice, evaluation was done in a 5-shot fashion with random examples. In all cases, we computed the log probabilities of all candidates, and picked the one with the highest score.

**Commercial Models.** We kept the evaluation as similar as possible to allow a fair comparison with open models. We used the same prompts and provided few-shot examples as user and assistant messages. In addition, we used the following system prompt in English to specify the expected answer format: `Respond always with a single letter: A, B or C`. All experiments with closed models were performed using the official APIs from OpenAI and Anthropic.

## 4 Results

In this section, we present the results and findings from our experiments. We first report the main results on the English version of BERTAQA, revealing that existing models struggle with local knowledge (Section 4.1). Section 4.2 shows that continued pretraining in Basque can improve performance on local questions, proving that knowledge can be transferred from a low-resource to a high-resource language. Section 4.3 reports results on the Basque version of the benchmark, revealing that existing models fail to encode knowledge in a fully language-agnostic manner. Finally, Section 4.4 covers translate-test and self-translate, showing that they are less effective for local questions. Appendices E and F report additional results by category and difficulty.

### 4.1 Main results

We first evaluate existing models in English, focusing on the difference in performance between local and global topics. As shown in Table 2, most models obtain good results on global questions. However, the performance consistently drops when evaluated on local topics, with a gap of 26.66 points on average. More concretely, state-of-the-art models like GPT-4 Turbo and Claude 3 Opus shine on the global subset, scoring above 90% accuracy. Nevertheless, these same models obtain about 72% accuracy on the local subset. The difference is even larger for open-weight models, with the best one (Llama 3 70B) obtaining less than 60% accuracy on local questions. This confirms that, despite the impressive performance of LLMs in knowledge-intensive tasks, subjects pertinent to minority cultures remain challenging. As reported in Appendices E and F, results further drop to around 53% for the worst category and 66% for the hardest difficulty, leaving ample room for improvement for future work. We manually inspected a subset of 50 local questions that the best models (GPT 4 Turbo and Claude 3 Opus) answered incorrectly in Section 4.5.

Table 2: Results for the English version of BERTAQA. The $\Delta$ column shows the difference between local and global results. Best results and smallest $\Delta$ differences are in bold.

| Model | Variant | Local | Global | $\Delta$ |
|---|---|---|---|---|
| Random | N/A | 33.33 | 33.33 | 0.00 |
| GPT | 3.5 Turbo | 55.08 | 82.40 | 27.32 |
| | 4 | 69.88 | 91.43 | 21.55 |
| | 4 Turbo | **72.17** | **91.68** | **19.51** |
| Claude 3 | Haiku | 58.71 | 84.16 | 25.45 |
| | Sonnet | 58.33 | 86.41 | 28.08 |
| | Opus | **71.91** | **91.85** | **19.94** |
| Llama 2 | 7B | 41.54 | 64.34 | **22.80** |
| | 13B | 43.61 | 70.36 | 26.75 |
| | 70B | **49.15** | **77.68** | 28.53 |
| Llama 3 | 8B | 50.38 | 76.63 | 26.25 |
| | 70B | **59.56** | **84.74** | **25.18** |
| Qwen 1.5 | 7B | 42.51 | 71.45 | **28.94** |
| | 14B | 44.67 | 75.92 | 31.25 |
| | 72B | **54.70** | **83.99** | 29.29 |
| Yi | 6B | 44.25 | 73.20 | **28.95** |
| | 9B | 43.87 | 75.00 | 31.13 |
| | 34B | **54.06** | **83.61** | 29.55 |
| Mistral | 7B | 47.50 | 74.16 | 26.66 |
| | 47B | **57.40** | **82.78** | **25.38** |
| Gemma | 7B | **45.69** | **76.42** | **30.73** |
| **Average** | N/A | 53.25 | 79.91 | 26.66 |

Table 3: Effect of continually pretraining Llama 2 in Basque on the English version of BERTAQA. The best results for each size and group are in bold.

| Model | Local | Global | $\Delta$ |
|---|---|---|---|
| Llama 2 7B | 41.54 | **64.34** | 22.80 |
| + *eu train* | **47.72** | 53.26 | **5.54** |
| Llama 2 13B | 43.61 | **70.36** | 26.75 |
| + *eu train* | **56.60** | 67.47 | **10.87** |
| Llama 2 70B | 49.15 | **77.68** | 28.53 |
| + *eu train* | **62.61** | 73.62 | **11.01** |

Interestingly, we find that the performance on local and global questions is strongly correlated for the models we tested (the Pearson correlation between the two scores is 0.844). Models obtaining a similar score on the local subset also obtain a similar score on the global subset. We presume that, if the training corpus of a given model was significantly more skewed towards local topics than that of another model, the former would tend to perform better in local topics, at least in relative terms. Given that we do not observe this, we hypothesize that the training recipes of existing models are roughly equivalent in how they balance global and local knowledge. However, we do find notable differences on how scaling impacts local vs. global questions for different models. For model families with the lowest scores in BertaQA, like Llama 2, scaling yields bigger gains on the global subset, as the delta between local and global questions increases from 22.80 for the smallest variant to 28.53 for the largest variant. The opposite is true for more performant model families like GPT, with the delta between local and global questions going from 27.32 for GPT-3.5 Turbo to 19.51 for GPT-4 Turbo. This suggests that it is generally easier to improve on global questions, but this subset starts saturating for the strongest models, resulting in bigger improvements on the local subset.

## 4.2 Local knowledge transfer from Basque to English

As we have just seen, existing LLMs perform much better on global questions compared to local questions. One possible explanation is that their training data is dominated by English, which has become the de-facto world language, capturing extensive knowledge about global subjects. However, knowledge about other cultures can be scarce in English when the corresponding community speaks a different language. For instance, English Wikipedia is considerably bigger than Basque Wikipedia, but articles about Basque traditions, literature or music tend to be more extensive in the Basque language. For that reason, we hypothesize that effectively leveraging training corpora in these other languages can help bridge the gap between local and global knowledge.

To test this hypothesis, we experiment with Latxa [Etxaniz et al., 2024], a family of Basque base language models that were built by continuing training Llama 2 on Basque corpora. Latxa was trained on all publicly available corpora in Basque meeting some minimum quality standards. This mostly corresponds to crawling corpora, including both processed versions of CommonCrawl as well as ad-hoc crawling of websites with high-quality content. The largest portion of it consists of news, which is the most common use of Basque on the web. While the resulting pretraining corpus is diverse in nature, we would not say that the domains and topics in BertaQA are particularly prominent on it, although it is obviously more likely that topics related to the Basque culture are discussed in the Basque language compared to English. As shown in Table 3, this continued training in Basque brings large improvements on the local subset, outperforming the original Llama 2 model by 13.46 points in the case of the 70B variant. It is remarkable that we observe these gains when performing evaluation in English, even if the continued training is done in Basque, which implies that there is knowledge transfer from Basque into English. This challenges the conventional wisdom that adding more languages hurts English performance, a phenomenon known as the *curse of multilinguality* [Conneau et al., 2020, Pfeiffer et al., 2022, Chang et al., 2024]. To the best of our knowledge, this is the first solid evidence of knowledge being transferred from a low-resource to a high-resource language.

Nevertheless, we observe the opposite effect on the global subset, where the continued training in Basque hurts performance. The degradation is relatively small for the 13B and 70B models, but more notable for the 7B model. This suggests that training on Basque data improves English performance on subjects related to Basque culture, while hurting performance on more general topics. Given that prior work mostly evaluated on global subjects, this led to the generally accepted conclusion that training on other languages harms English. We show that this conclusion does not necessarily show the full picture, since models have barely been evaluated on local topics, and their behavior there can be fundamentally different.

## 4.3 Comparison of English and Basque results

All of our results so far correspond to the English version of BERTAQA. In this section, we focus on the Basque version instead. Recall that the English and Basque versions are parallel (i.e. they consist of the exact same questions in different languages), so the gap in performance between the two variants reflects how effective LLMs are at leveraging the same knowledge in each language.

As shown in Table 4, the vast majority of models obtain worse results in Basque, both on local and global topics. This is expected, as these models were primarily trained in English. Despite this, many models remain competitive on the global subset, demonstrating that many questions can be answered even with limited knowledge of Basque.

The only exception is the extension of Llama 2 with continued training in Basque (*Llama 2 + eu train*). For this model, the best local results are obtained in Basque, whereas the best global results are obtained in English. This implies that the previously observed knowledge transfer between Basque and English (Section 4.2) is not perfect: while the continued training in Basque did improve local performance in English, the model performs even better in Basque. Similarly, the global knowledge coming from Llama 2 does not transfer completely to Basque. This suggests that LLMs fail to encode knowledge in a completely language-agnostic manner, and tend to perform better when queried in the original language that they acquired the relevant knowledge in.

Table 4: Results for the Basque version of BERTAQA. The $\Delta$ column shows the difference between local and global results. Numbers in parentheses show the differences with the English results. Best results and smallest $\Delta$ differences are in bold.

| Model | Variant | Local | Global | $\Delta$ |
|---|---|---|---|---|
| Random | N/A | 33.33 | 33.33 | 0.00 |
| GPT | 3.5 Turbo | 47.25 (-7.83) | 66.22 (-16.18) | **18.97** |
| | 4 | 62.94 (-6.94) | 85.91 (-5.52) | 22.97 |
| | 4 Turbo | **69.46** (-2.71) | **89.21** (-2.47) | 19.75 |
| Claude 3 | Haiku | 58.21 (-0.50) | 79.85 (-4.31) | 21.64 |
| | Sonnet | 56.13 (-2.20) | 83.24 (-3.17) | 27.11 |
| | Opus | **71.32** (-0.59) | **90.89** (-0.96) | **19.57** |
| Llama 2 | 7B | 34.90 (-6.64) | 37.08 (-27.26) | **2.18** |
| | 13B | 34.09 (-9.52) | 43.77 (-26.59) | 9.68 |
| | 70B | **37.39** (-11.76) | **54.22** (-23.46) | 16.83 |
| Llama 2 + *eu train* | 7B | 49.45 (+1.73) | 50.79 (-2.47) | **1.34** |
| | 13B | 60.24 (+3.64) | 65.47 (-2.00) | 5.23 |
| | 70B | **64.85** (+2.24) | **72.24** (-1.38) | 7.39 |
| Llama 3 | 8B | 42.60 (-7.78) | 63.09 (-13.54) | **20.49** |
| | 70B | **57.40** (-2.16) | **82.15** (-2.59) | 24.75 |
| Qwen 1.5 | 7B | 35.96 (-6.55) | 46.15 (-25.30) | **10.19** |
| | 14B | 37.31 (-7.36) | 53.39 (-22.53) | 16.08 |
| | 72B | **42.77** (-11.93) | **63.25** (-20.74) | 20.48 |
| Yi | 6B | 37.94 (-10.32) | 46.45 (-22.99) | **8.51** |
| | 9B | 38.20 (-13.79) | 49.21 (-21.70) | 11.01 |
| | 34B | **41.03** (-6.31) | **60.41** (-26.75) | 19.38 |
| Mistral | 7B | 37.18 (-5.67) | 51.17 (-25.79) | **13.99** |
| | 47B | **43.61** (-13.03) | **61.08** (-23.20) | 17.47 |
| Gemma | 7B | **41.84** (-3.85) | **65.89** (-10.53) | 24.05 |
| **Average** | N/A | 47.92 (-5.64) | 63.53 (-14.41) | 15.61 |

## 4.4 Translate-test and self-translate

Translating the test data into English is a popular approach to performing cross-lingual learning in low-resource languages [Ahuja et al., 2023]. In this section, we compare how existing translation-based approaches behave on local and global topics. Specifically, we experiment with two methods: *translate-test*, where the Basque input is translated into English using an external machine translation system (NLLB-200; [Costa-jussà et al., 2022]), and *self-translate*, where the LLM itself produces the English translation in a few-shot fashion [Etxaniz et al., 2023].

As shown in Table 5, translation-based approaches tend to be more effective for global questions. The best example is Gemma, for which both translate-test and self-translate improve performance on the global subset, while harming performance on the local subset. For Llama 2, translating into English is almost always helpful, which is not surprising as this model is not versed in Basque. However, the improvements are substantially larger for global questions. In contrast, translation approaches are generally harmful for the extension of Llama 2 with continued training in Basque although, once again, the degradation is smaller for global questions. Together, this suggests that the positive results for translate-test and self-translate in prior work might be inflated by the fact that their evaluation was limited to global topics.

## 4.5 Error Analysis

As we stated previously, some local questions are still challenging even for the best models. To get more insights of the nature of these questions, we do a qualitative analysis of of 50 local questions in English that the best models (GPT 4 Turbo and Claude 3 Opus) answered incorrectly. As expected, most questions in this sample are of medium or hard difficulty, and fall in the most challenging

Table 5: Results for translate-test and self-translate settings. Numbers in parentheses show the difference with direct inference in Basque. Best results are in bold.

| Model | Size | Method | Local | Global |
|---|---|---|---|---|
| Llama 2 | 7B | Translate-test | **37.44** (+2.54) | **55.35** (+18.27) |
| | | Self-translate | 33.80 (-1.10) | 38.71 (+1.63) |
| | 13B | Translate-test | **37.69** (+3.60) | **62.50** (+18.73) |
| | | Self-translate | 34.81 (+0.72) | 46.11 (+2.34) |
| | 70B | Translate-test | **42.68** (+5.29) | **71.03** (+16.81) |
| | | Self-translate | 39.85 (+2.46) | 55.23 (+1.01) |
| Llama 2 + *eu train* | 7B | Translate-test | 35.79 (-13.66) | 44.27 (-6.52) |
| | | Self-translate | **44.37** (-5.08) | **50.04** (-0.75) |
| | 13B | Translate-test | 41.79 (-18.45) | 59.36 (-6.11) |
| | | Self-translate | **56.13** (-4.11) | **65.55** (+0.08) |
| | 70B | Translate-test | 46.28 (-18.57) | 65.47 (-6.77) |
| | | Self-translate | **60.15** (-4.70) | **70.48** (-1.76) |
| Gemma | 7B | Translate-test | **41.67** (-0.17) | **69.19** (+3.30) |
| | | Self-translate | **41.67** (-0.17) | 67.68 (+1.79) |

categories. None of the questions can be answered by common sense, and distractors are generally challenging.

Next, we try to find the correct answer of these questions by searching the web, to measure how difficult they can be. We include half of the examples in Table 6, the rest can be found in Appendix G. First, we found the correct answer relatively easily in 24 of the questions. Next, in 20 questions, finding the correct answer was more challenging, requiring multiple web searches, or searching the web in Basque. Six of these answers where trickier to find in English, often leading to no answer or even incorrect answers. Finally, in 16 of the questions we were unable to find the correct answer. In half of the questions, searching the web led to the wrong answer. Some of these questions involve temporal variations, the correct answer has changed in time. There were also some questions where we found no answer, such as the ones including negation.

## 5 Related Work

Research in NLP evaluation has predominantly focused in English, with most multilingual benchmarks being translated from this language, such as XNLI [Conneau et al., 2018], XQUAD [Artetxe et al., 2019], MLQA [Lewis et al., 2019] and Belebele [Bandarkar et al., 2023]. This parallel nature facilitates monolingual, multilingual, and cross-lingual experiments, enabling valuable comparisons across languages. However, this approach introduces biases related to translations and cultural representation, affecting experimental conclusions by reflecting the culture of the original dataset.

Recently, there has been a focus on creating native evaluation benchmarks to assess local cultural knowledge, rather than relying on translations from English. These native datasets, which resemble popular English benchmarks, include unique cultural elements that are generally more challenging for current models. They usually are of higher quality than machine or human-translated datasets. For example, native MMLU [Hendrycks et al., 2020] datasets have been created for Chinese [Li et al., 2023], Korean [Son et al., 2024], Indonesian [Koto et al., 2023] and Arabic [Koto et al., 2024]. Other examples of language-specific evaluation benchmarks include C-Eval for Chinese [Huang et al., 2024], HAE-RAE Bench for Korean [Son et al., 2023], COPAL-ID for Indonesian [Wibowo et al., 2023] and RoCulturaBench for Romanian [Masala et al., 2024]. Finally, Etxaniz et al. [2024] introduces 4 native Basque multiple-choice evaluation datasets that include local questions.

Another relevant benchmark is SeaEval [Wang et al., 2023], which introduces 4 datasets for multi-cultural reasoning and 2 for cross-lingual consistency. The multicultural datasets include various countries and languages: the United States (English), Singapore (English), China (Chinese), and the Philippines (English). The cross-lingual consistency dataset covers common knowledge in 7 diverse languages: English, Chinese, Indonesian, Spanish, Vietnamese, Malay, and Filipino.

Table 6: First 25 error analysis examples annotated by difficulty of web search.

| Category | Diff. | Question | Candidate 0 | Candidate 1 | Candidate 2 | Ans. | Web |
|---|---|---|---|---|---|---|---|
| Basque and Literature | 1 | What does the "Karmel" magazine specialize in? | Bertsolarism | Basque culture in the past and the present | The life of the Carmelites | 1 | diff. |
| Geography and History | 3 | Which of these towns does not have a common boundary with Elorrio? | Zaldibar | Bergara | Matiena | 2 | easy |
| Geography and History | 3 | Which of these districts is not in the Pettarra area of Soule? | Etxarri | Garruze | Sarricotapea | 1 | no |
| Sports and Leisure | 2 | What's Aritz Aranburu's birthplace? | Zarautz | Orio | Getaria | 2 | easy |
| Basque and Literature | 2 | Which Basque dialect is spoken in Eibar? | The Lower Deba dialect | The Central dialect | The Western dialect | 2 | easy |
| Basque and Literature | 2 | Which of these three was the Head of a Department in the Chartered Provincial Council of Gipuzkoa? | Tere Irastortza | Joan Mari Irigoien | Xabier Lete | 2 | diff. |
| Cinema and Shows | 2 | The stories of how many young people is the "Hasiberriak" TV show about? | Ten | Seven | Five | 0 | diff. |
| Geography and History | 1 | Which is the longest river that flows into the sea in the Basque Country? | The Nervion | The Adour | The Ebro | 1 | diff. |
| Geography and History | 2 | Which of these is the least populated area? | The canton of Maule/Mauleón | The canton of Donapaleu/ Saint-Palais | The canton of Atharratze/ Tardets | 1 | diff. |
| Science and Technology | 2 | How many echidna species are there? | 8 | 5 | 3 | 1 | no |
| Basque and Literature | 1 | What is Ricardo Arregi Diaz de Heredia mostly involved in? | Journalism | Adventure stories | Poetry | 2 | easy |
| Geography and History | 3 | What year was "La Vizcaya" factory set up? | In 1882 | In 1884 | In 1886 | 0 | no |
| Geography and History | 1 | What century does the oldest Basque text we know written on paper belong to? | The 15th century | The 12th century | The 10th century | 2 | no |
| Music and Dance | 1 | Where was the great dance master Iñaki Irigoien from? | From Bilbao | From Donostia / San Sebastian | From Vitoria-Gasteiz | 0 | easy |
| Geography and History | 3 | Which king of Navarre died in 882? | Fortun Gartzia | Eneko Aritza | Gartzia Eneko | 2 | difficult |
| Cinema and Shows | 2 | Which band wrote the opening song of the "Pilotari" TV series? | Gari | Sugan | Ken Zazpi | 1 | difficult |
| Basque and Literature | 2 | Which of these subjects was not addressed at the Basque Floral Games? | Dance | Rural sports | Basque poetry | 1 | no |
| Sports and Leisure | 3 | When was the Leurtza reservoir built? | In 1920 | In 1925 | In 1921 | 0 | easy |
| Science and Technology | 2 | Where's the Basque Museum of Medical History? | In Bilbao | In Leioa | In Barakaldo | 1 | easy |
| Music and Dance | 2 | Where does the "Axuri Beltza" dance come from? | From the Biscayan town of Aulestia | From the Gipuzkoan town of Errezil | From the Navarrese town of Jaurrieta | 2 | difficult |
| Cinema and Shows | 3 | When is the "Teknopolis" programme broadcast on ETB1 (Basque Public TV)? | From Monday to Thursday at 18:00 | On Saturdays at 15:00 | On Fridays and Saturdays at 11:00 | 1 | no |
| Science and Technology | 2 | What is the other name of the first blast furnace of Altos Hornos de Vizcaya? | The Miren Agote furnace | The Maria Angeles furnace | The Santa Ana furnace | 1 | no |
| Music and Dance | 3 | How many dance championships are held in the Northern Basque Country a year? | None | Three | Five | 0 | no |
| Music and Dance | 2 | Where does the "Trapatan" dance take place today? | In Etxarri Aranatz | In Ituren | In Doneztebe/Santesteban | 2 | difficult |
| Cinema and Shows | 2 | How many Basque voices took part in the show "Sortuko dira besteak"? | Six | Eight | Four | 0 | difficult |

Analysis of the cultural bias of LLMs has attracted some interest in recent years. Havaldar et al. [2023] concluded that multilingual models are not multicultural, whereas Tao et al. [2023] found that GPT-4, 3.5 and 3 exhibit cultural values resembling English-speaking and Protestant European countries. In a similar vein, Naous et al. [2023] also found that multilingual and Arabic monolingual LMs exhibit bias towards Western culture.

According to Liu et al. [2024] translating into English can improve the performance of English-centric LLMs on most multilingual tasks. However, for culturally related tasks requiring deeper language understanding, prompting in the native language proves to be more effective since it can capture the nuances related to culture and language. This aligns with our findings: Latxa (dubbed "+ eu train" in the experimental sections) performs better in Basque for local topics, and better in English for global topics. On the other hand, AlKhamissi et al. [2024] found that these models exhibit a higher degree of cultural alignment when they are prompted with the predominant language of the culture, and have been pre-trained on the main language of the culture. In our case, empirical results also show that pretraining in Basque improves Basque culture knowledge, and prompting in Basque leads to better results than English.

The study of LLMs from a cultural perspective is challenging. Adilazuarda et al. [2024] observed, after a survey of 39 recent papers, that none of the studies define "culture", which is a complex, multifaceted concept. Instead, they probe models on some datasets which represent certain aspects of "culture", leaving other aspects untested. Ramesh et al. [2023] argue that the vast array of cultures and languages worldwide makes it impractical to create datasets that cover all. As a result, they believe that identifying and addressing biases should move away from relying only on datasets that have limited reach and are not adaptable to every language and culture.

Despite the extensive related work on the topic, our new benchmark is unique because it is natively created, provides a professionally translated English version, and distinguishes between local and global questions. Other datasets may include local questions, but they lack specific annotations to separate them from global ones. This distinction in our dataset enables more precise experiments to analyze the limitations of models.

## 6   Conclusion and Future Work

Most existing NLP benchmarks are limited to anglocentric or global subjects. So as to understand how LLMs perform on subjects that are idiosyncratic to other cultures, we introduce BERTAQA, a trivia dataset comprising a local subset with questions about the Basque culture, and a global subset with questions of broader interest. Our results show that state-of-the-art models struggle with local knowledge, despite excelling on global subjects. In addition, we find that some prior findings need to be reconsidered when reassessed on local topics. In particular, we find that continued pretraining can transfer local knowledge from Basque into English, challenging the conventional wisdom that training on low-resource languages harms high-resource languages. In addition, we show that translation-based techniques like translate-test and self-translate are more effective on global questions, suggesting that results in prior work were inflated. Given that we often observe diverging trends in local and global questions, we believe that it is critical to cover them both when evaluating LLMs in the future.

While our work is a first step in this direction, it also comes with some limitations, which we would like to address in future work. More concretely, the local subset of our benchmark is limited to questions about the Basque culture. While we expect that the general trends we observe would also apply to other minority cultures, we believe that it would be valuable to build similar datasets covering other cultures. This is generally more challenging than developing benchmarks about global subjects, as it usually requires being part of or engaging with the relevant communities. We hope that our work prompts to reconsider how LLMs are evaluated more broadly, and motivates the creation of similar datasets for other minority cultures. Besides, the English version of our dataset was created through professional translation, which could lead to *translationese* and other translation artifacts [Artetxe et al., 2020]. This typically occurs in the opposite direction, as most datasets are translated from English into other languages. In the future, we would like to analyze if this has any impact in our results, and explore authoring questions about minority cultures in English rather than translating from their respective languages.

## Acknowledgements

Julen is funded by a PhD grant from the Basque Government (PRE_2023_2_0060). This work is partially supported by projects DeepR3 TED2021-130295B-C31, and DeepKnowledge PID2021-127777OB-C21 funded by MCIN/AEI/10.13039/501100011033 and European Union NextGeneration EU/PRTR, as well as and the Basque Government (IXA excellence research group IT1570-22, IKER-GAITU 11:4711:23:410:23/0808) and the Spanish Ministry for Digital Transformation and of Civil Service, and the EU-funded NextGenerationEU Recovery, Transformation and Resilience Plan (ILENIA project, 2022/TL22/00215335).

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

# A  Statistics

We show detailed statistics of each category, group and difficulty in Table 7.

Table 7: Statistics by Category and Group. We show the number of total items and items per difficulty. We also report the average number of characters of questions and candidate answers in English and Basque.

| Category | Group | Items | Difficulty | | | English chars | | Basque chars | |
|---|---|---|---|---|---|---|---|---|---|
| | | | Easy | Medium | Hard | Question | Candidates | Question | Candidates |
| Basque and Literature | Local | 305 | 90 | 103 | 112 | 55.9 | 16.9 | 51.0 | 16.7 |
| Basque and Literature | Global | 310 | 91 | 108 | 111 | 53.3 | 15.7 | 51.1 | 16.0 |
| Geography and History | Local | 300 | 110 | 110 | 80 | 48.9 | 12.8 | 44.4 | 12.1 |
| Geography and History | Global | 300 | 110 | 110 | 80 | 43.0 | 10.3 | 43.7 | 11.0 |
| Society and Traditions | Local | 289 | 103 | 108 | 78 | 60.2 | 16.1 | 53.7 | 14.9 |
| Society and Traditions | Global | 298 | 110 | 109 | 79 | 51.1 | 18.0 | 50.4 | 18.6 |
| Sports and Leisure | Local | 296 | 107 | 109 | 80 | 47.5 | 11.7 | 42.6 | 10.7 |
| Sports and Leisure | Global | 303 | 113 | 110 | 80 | 43.3 | 10.3 | 43.0 | 10.5 |
| Culture and Art | Local | 295 | 105 | 110 | 80 | 43.5 | 11.5 | 39.5 | 10.2 |
| Culture and Art | Global | 286 | 98 | 108 | 80 | 40.7 | 9.1 | 38.1 | 9.6 |
| Music and Dance | Local | 289 | 107 | 102 | 80 | 49.0 | 12.5 | 45.8 | 13.0 |
| Music and Dance | Global | 300 | 110 | 110 | 80 | 43.4 | 10.8 | 41.6 | 11.6 |
| Science and Technology | Local | 292 | 105 | 108 | 79 | 63.3 | 12.5 | 60.0 | 12.7 |
| Science and Technology | Global | 296 | 108 | 109 | 79 | 53.0 | 11.0 | 54.0 | 11.3 |
| Cinema and Shows | Local | 298 | 110 | 109 | 79 | 67.1 | 16.3 | 65.2 | 16.7 |
| Cinema and Shows | Global | 299 | 109 | 110 | 80 | 55.8 | 12.4 | 59.3 | 13.5 |
| All | Local | 2364 | 837 | 859 | 668 | 54.4 | 13.8 | 49.0 | 13.4 |
| All | Global | 2392 | 849 | 874 | 669 | 48.0 | 12.2 | 47.7 | 12.8 |
| All | All | 4756 | 1686 | 1733 | 1337 | 51.2 | 13.0 | 49.0 | 13.1 |

# B  Basque Examples

The same examples that were included in English in Table 1 are included in Basque in Table 8.

# C  Prompts

Regarding the prompts, we used the Basque prompts described in Etxaniz et al. [2024] for multiple choice questions, which were manually translated for the English experiments (see Table 9. The answer `choices` were single letters (A, B, C) and the `answer` index was used as the index of the correct answer.

# D  Compute

The experiments we performed are not very compute-intensive. We performed all the experiments using A100 80GB GPUs in our internal cluster. The largest models require using at least 3 GPUs. Evaluating each model took a few minutes, so the total compute is of a few GPU hours.

# E  Results by Category

We show that previous local and global results are consistent across categories in Tables 10 and 11. The large difference between local and global is maintained across all models and categories. When comparing Llama 2 and Latxa, we see that previous results are consistent across all categories. That is, Latxa is better at local questions and Llama 2 is better at global questions. However, we see that the differences vary significantly depending on the models and categories.

For example, Latxa obtains very good results in the Basque and Literature category, both in local and global questions. In local questions, it is on par with the best commercial models. These results

Table 8: Basque examples of local and global questions in each category.

| | Local Questions | Global Questions |
|---|---|---|
| **Basque and Literature** | Zertaz ari da "Karmel" aldizkaria? 
 a) Bertsolaritzaz 
 **b) Lehengo eta gaurko euskal kulturaz** 
 c) Karmeldarren bizimoduaz | Eleberri hauetako zeinetan ez da agertzen itsasoa? 
 **a) "Tom Sawyer-ren abenturak"** 
 b) "Moby Dick" 
 c) "Altxorraren uhartea" |
| **Geography and History** | Non dago Atxondo? 
 **a) Bizkaian** 
 b) Gipuzkoan 
 c) Nafarroan | Nor kartzelaratu zuten 1964an? 
 **a) Nelson Mandela** 
 b) Mumia Abu Jamal 
 c) Charles Ghankay |
| **Society and Tradition** | Hauetako zein dago Eusko Jaurlaritzaren menpe? 
 a) IKA 
 b) AEK 
 **c) HABE** | Zein da gehien erabiltzen dugun energia mota? 
 **a) Petrolioa** 
 b) Hidroelektrikoa 
 c) Energia nuklearra |
| **Sports and Leisure** | Non jaio zen Julian Retegi? 

 a) Areson 
 **b) Eratsunen** 
 c) Erason | Nork irabazi ditu munduko futbol-txapelketa gehien? 
 a) Argentinak 
 b) Alemaniak 
 **c) Brasilek** |
| **Culture and Art** | Nortzuek eraiki zuten Baionako Gaztelu Berria? 
 **a) Ingelesek** 
 b) Frantsesek 
 c) Espainolek | Noiz ireki zuten Titanic Belfast Museoa? 
 **a) 2012an** 
 b) 2005ean 
 c) 2002an |
| **Music and Dance** | Nongo dantza da jatorriz "dantzari" izeneko dantza? 
 a) Busturialdekoa 
 b) Enkarterrietakoa 
 **c) Durangaldekoa** | Nork idatzi zuen James Bond serieko soinu-banda? 
 **a) John Barry-k** 
 b) Henry Mancini-k 
 c) John Williams-ek |
| **Science and Technology** | Bizkaiko zer udalerri dago lotuta dinamitarekin? 
 a) Leioa 
 **b) Galdakao** 
 c) Erandio | Zein da daltonismoaren izen zientifikoa? 
 a) Chondrostoma 
 b) Kromotosia 
 **c) Diskromatopsia** |
| **Cinema and Shows** | Nola du izena Bernardo Atxagaren "Obabakoak" ele-berrian oinarritutako filmak? 

 a) "Obabakoak" 
 **b) "Obaba"** 
 c) "Obabako istorioak" | Some like it hot filmean (gaztelaniaz, "Con faldas y a lo loco"), zer instrumentu jotzen zuen Marilyn Monroek? 
 a) Arpa 
 b) Didgeridoo 
 **c) Ukelele** |

Table 9: Prompts

| **Basque** | **English** |
|---|---|
| Galdera: {question} 
 A. {candidates[0]} 
 B. {candidates[1]} 
 C. {candidates[2]} 
 Erantzuna: {answer} | Question: {question} 
 A. {candidates[0]} 
 B. {candidates[1]} 
 C. {candidates[2]} 
 Answer: {answer} |

match previous results in Latxa [Etxaniz et al., 2024], where Latxa surpasses the best commercial models on language proficiency and trivia questions in the Language and Literature categories.

There are more categories where Latxa is on par with the best models on local topics, such as Music and Dance and Cinema and Shows. These categories are also the ones where the best models struggle the most on local questions, and the difference with global topics is the largest.

# F    Results by Difficulty

Results are also consistent across difficulty levels, as shown in Tables 10 and 13. All models obtain worse results in local questions at all difficulty levels. There is a clear difference between difficulties, with the biggest drop in performance happening from the easy to medium difficulty. Commercial models have a bigger drop in scores on local questions than on global questions for medium and hard

Table 10: Results of models in English by category and group.

| Model | Variant | Basque and Literature | | Geography and History | | Society and Traditions | | Sports and Leisure | | Culture and Art | | Music and Dance | | Science and Technology | | Cinema and Shows | |
|---|---|---|---|---|---|---|---|---|---|---|---|---|---|---|---|---|---|
| | | Loc | Glo | Loc | Glo | Loc | Glo | Loc | Glo | Loc | Glo | Loc | Glo | Loc | Glo | Loc | Glo |
| GPT | 3.5 Turbo | 51.5 | 74.5 | 56.7 | 85.0 | 64.4 | 85.2 | 55.1 | 81.9 | 53.6 | 84.3 | 42.2 | 78.0 | 62.3 | 84.5 | 55.0 | 86.3 |
| | 4 | 67.9 | 85.5 | 70.7 | 92.0 | 81.7 | 92.6 | 65.5 | 91.8 | 72.2 | 95.8 | 54.3 | 91.0 | 80.8 | 89.9 | 66.1 | 93.3 |
| | 4 Turbo | 75.4 | 86.1 | 77.7 | 90.0 | 83.0 | 92.3 | 70.3 | 93.1 | 76.6 | 96.9 | 52.9 | 89.7 | 76.0 | 90.9 | 65.1 | 95.0 |
| Claude 3 | Haiku | 65.6 | 79.0 | 62.0 | 85.7 | 67.5 | 88.9 | 54.1 | 83.5 | 55.3 | 86.0 | 43.6 | 76.3 | 69.2 | 86.8 | 52.4 | 87.3 |
| | Sonnet | 60.3 | 79.4 | 62.3 | 85.7 | 66.8 | 90.3 | 55.4 | 87.8 | 57.3 | 93.0 | 37.0 | 78.7 | 70.9 | 89.5 | 56.4 | 87.6 |
| | Opus | 77.7 | 89.0 | 69.3 | 92.7 | 85.5 | 93.3 | 68.2 | 91.4 | 76.6 | 96.5 | 49.8 | 86.3 | 79.8 | 91.2 | 68.1 | 94.7 |
| Llama 2 | 7B | 43.6 | 59.0 | 41.3 | 66.7 | 42.2 | 71.1 | 39.5 | 61.1 | 42.4 | 65.7 | 40.5 | 57.7 | 46.2 | 65.2 | 36.6 | 68.6 |
| | 13B | 41.0 | 57.4 | 43.3 | 76.0 | 48.1 | 76.9 | 42.6 | 69.0 | 43.4 | 72.0 | 35.0 | 64.3 | 51.7 | 73.7 | 44.0 | 74.3 |
| | 70B | 45.6 | 67.4 | 54.3 | 82.3 | 57.1 | 86.2 | 47.6 | 75.9 | 48.5 | 76.6 | 40.5 | 67.3 | 55.8 | 83.1 | 44.0 | 82.9 |
| Latxa | 7B | 51.2 | 54.2 | 43.7 | 55.3 | 52.6 | 53.4 | 47.6 | 49.2 | 45.8 | 54.2 | 47.8 | 48.3 | 44.2 | 58.1 | 49.0 | 53.5 |
| | 13B | 66.2 | 71.3 | 53.3 | 72.0 | 64.0 | 70.5 | 51.4 | 65.4 | 56.6 | 65.4 | 50.9 | 59.7 | 57.2 | 67.9 | 53.0 | 67.6 |
| | 70B | 76.4 | 77.1 | 58.0 | 74.0 | 70.6 | 79.5 | 54.1 | 71.0 | 58.0 | 73.8 | 50.2 | 60.7 | 67.5 | 79.7 | 65.8 | 73.2 |
| Llama 3 | 7B | 48.2 | 61.6 | 46.3 | 69.0 | 40.5 | 61.7 | 39.2 | 55.5 | 44.8 | 66.8 | 36.0 | 57.3 | 44.2 | 68.2 | 41.3 | 64.9 |
| | 70B | 60.7 | 74.8 | 65.3 | 84.3 | 65.1 | 87.9 | 51.7 | 81.5 | 53.9 | 85.3 | 37.4 | 76.3 | 68.2 | 85.8 | 56.7 | 81.6 |
| Qwen 1.5 | 7B | 46.2 | 60.0 | 40.0 | 74.7 | 40.8 | 74.8 | 38.2 | 70.0 | 41.0 | 77.6 | 36.3 | 63.3 | 51.0 | 78.0 | 46.3 | 73.9 |
| | 14B | 43.6 | 62.9 | 41.0 | 79.0 | 47.1 | 83.6 | 42.9 | 76.2 | 45.4 | 81.8 | 38.1 | 66.3 | 55.5 | 81.8 | 44.0 | 76.6 |
| | 72B | 50.5 | 72.3 | 56.3 | 86.0 | 64.7 | 88.9 | 52.0 | 83.5 | 54.9 | 88.8 | 40.8 | 76.3 | 67.1 | 87.5 | 51.3 | 89.3 |
| Yi | 6B | 39.3 | 59.7 | 46.3 | 77.0 | 48.4 | 80.9 | 42.6 | 72.0 | 43.4 | 76.2 | 36.0 | 65.3 | 50.0 | 78.0 | 48.0 | 77.3 |
| | 9B | 40.3 | 67.1 | 42.3 | 78.7 | 44.3 | 81.2 | 42.2 | 78.2 | 44.1 | 77.6 | 35.6 | 61.0 | 52.7 | 82.8 | 49.3 | 73.9 |
| | 34B | 48.9 | 71.0 | 56.0 | 87.7 | 62.3 | 88.9 | 50.7 | 83.5 | 55.9 | 90.2 | 40.5 | 73.7 | 64.7 | 88.2 | 53.7 | 86.6 |
| Mistral | 7B | 44.9 | 64.5 | 49.7 | 77.0 | 52.3 | 79.5 | 49.7 | 73.6 | 45.1 | 79.7 | 38.8 | 63.3 | 57.2 | 79.1 | 42.6 | 77.3 |
| | 47B | 59.7 | 73.2 | 59.7 | 84.3 | 63.7 | 87.3 | 55.1 | 83.2 | 55.6 | 86.0 | 42.6 | 76.3 | 67.5 | 87.2 | 55.4 | 85.3 |
| Gemma | 7B | 45.3 | 68.4 | 47.0 | 80.3 | 50.5 | 78.5 | 43.6 | 75.3 | 41.7 | 81.1 | 36.7 | 65.3 | 55.1 | 83.5 | 45.6 | 79.6 |
| Average | N/A | 54.3 | 70.2 | 54.0 | 79.8 | 59.3 | 81.5 | 50.4 | 76.2 | 52.7 | 80.5 | 41.9 | 69.7 | 60.6 | 80.9 | 51.7 | 79.6 |

Table 11: Results of models in Basque by category and group.

| Model | Variant | Basque and Literature | | Geography and History | | Society and Traditions | | Sports and Leisure | | Culture and Art | | Music and Dance | | Science and Technology | | Cinema and Shows | |
|---|---|---|---|---|---|---|---|---|---|---|---|---|---|---|---|---|---|
| | | Loc | Glo | Loc | Glo | Loc | Glo | Loc | Glo | Loc | Glo | Loc | Glo | Loc | Glo | Loc | Glo |
| GPT | 3.5 Turbo | 43.9 | 61.3 | 47.3 | 71.0 | 51.6 | 60.4 | 47.6 | 66.0 | 46.1 | 70.3 | 41.9 | 66.7 | 49.3 | 65.5 | 50.3 | 68.9 |
| | 4 | 63.0 | 81.0 | 64.0 | 87.3 | 76.5 | 88.6 | 60.8 | 87.5 | 66.8 | 89.5 | 49.1 | 79.7 | 63.7 | 83.5 | 59.7 | 90.6 |
| | 4 Turbo | 70.2 | 84.8 | 72.7 | 89.3 | 83.0 | 89.9 | 67.9 | 90.8 | 72.2 | 92.7 | 54.0 | 87.3 | 71.9 | 86.5 | 63.8 | 92.6 |
| Claude 3 | Haiku | 63.9 | 77.1 | 61.0 | 82.7 | 67.5 | 85.6 | 50.7 | 79.9 | 57.6 | 80.8 | 45.0 | 70.7 | 64.7 | 82.1 | 55.0 | 80.3 |
| | Sonnet | 62.3 | 77.7 | 56.3 | 89.0 | 65.1 | 85.2 | 52.7 | 83.5 | 55.9 | 87.1 | 40.5 | 74.3 | 66.1 | 86.8 | 50.0 | 82.6 |
| | Opus | 80.7 | 88.7 | 72.3 | 92.0 | 84.1 | 92.3 | 66.9 | 93.1 | 77.0 | 96.9 | 49.8 | 83.7 | 74.0 | 88.9 | 65.4 | 92.0 |
| Llama 2 | 7B | 35.4 | 37.7 | 34.7 | 38.0 | 36.7 | 37.9 | 38.2 | 32.0 | 33.6 | 43.4 | 31.1 | 33.0 | 36.6 | 33.1 | 32.9 | 41.8 |
| | 13B | 33.8 | 39.4 | 34.0 | 46.0 | 38.1 | 38.6 | 35.8 | 40.3 | 27.8 | 51.4 | 36.0 | 40.0 | 36.3 | 46.3 | 31.2 | 48.8 |
| | 70B | 38.7 | 51.9 | 36.0 | 61.3 | 36.3 | 51.3 | 37.2 | 47.9 | 40.0 | 58.4 | 35.3 | 51.3 | 38.0 | 53.4 | 37.6 | 58.5 |
| Latxa | 7B | 51.5 | 52.3 | 46.0 | 50.0 | 59.5 | 53.4 | 47.6 | 49.5 | 53.2 | 54.9 | 48.4 | 46.0 | 47.6 | 54.4 | 42.0 | 46.2 |
| | 13B | 72.1 | 71.3 | 57.3 | 71.0 | 72.3 | 66.8 | 53.4 | 62.1 | 61.7 | 65.0 | 49.8 | 53.3 | 59.3 | 65.5 | 55.7 | 68.6 |
| | 70B | 78.7 | 77.7 | 58.7 | 74.7 | 78.2 | 76.2 | 57.1 | 67.3 | 60.7 | 72.4 | 52.3 | 57.7 | 68.2 | 77.0 | 64.8 | 74.9 |
| Llama 3 | 8B | 48.2 | 61.6 | 46.3 | 69.0 | 40.5 | 61.7 | 39.2 | 55.5 | 44.8 | 66.8 | 36.0 | 57.3 | 44.2 | 68.2 | 41.3 | 64.9 |
| | 70B | 60.7 | 74.8 | 65.3 | 84.3 | 65.1 | 87.9 | 51.7 | 81.5 | 53.9 | 85.3 | 37.4 | 76.3 | 68.2 | 85.8 | 56.7 | 81.6 |
| Qwen 1.5 | 7B | 37.7 | 42.6 | 38.7 | 51.3 | 35.6 | 44.3 | 34.5 | 43.6 | 38.0 | 50.4 | 32.2 | 40.0 | 36.3 | 49.7 | 34.6 | 47.8 |
| | 14B | 41.0 | 49.4 | 37.0 | 60.3 | 34.3 | 50.0 | 35.8 | 49.8 | 40.7 | 57.7 | 35.0 | 46.3 | 36.6 | 53.7 | 37.9 | 60.2 |
| | 72B | 43.3 | 55.2 | 43.0 | 75.7 | 45.3 | 57.1 | 39.2 | 57.8 | 43.1 | 65.0 | 37.0 | 61.7 | 44.9 | 62.8 | 46.3 | 71.2 |
| Yi | 6B | 43.6 | 44.2 | 38.7 | 51.0 | 37.0 | 46.0 | 34.1 | 38.6 | 38.0 | 52.5 | 36.3 | 46.7 | 37.3 | 46.3 | 38.3 | 46.8 |
| | 9B | 38.7 | 44.5 | 35.3 | 57.3 | 37.7 | 43.6 | 36.5 | 43.9 | 42.4 | 51.1 | 35.6 | 44.0 | 40.1 | 53.4 | 39.3 | 56.2 |
| | 34B | 46.2 | 52.9 | 40.3 | 67.0 | 41.2 | 54.7 | 41.2 | 56.4 | 41.0 | 65.4 | 33.2 | 58.7 | 41.8 | 62.8 | 43.0 | 65.9 |
| Mistral | 7B | 37.7 | 47.1 | 35.0 | 51.7 | 40.8 | 49.7 | 36.2 | 46.2 | 38.3 | 58.0 | 33.6 | 45.3 | 42.8 | 52.0 | 33.2 | 59.9 |
| | 47B | 48.5 | 55.8 | 41.3 | 69.3 | 46.0 | 52.0 | 42.2 | 58.1 | 45.8 | 61.9 | 38.4 | 58.7 | 47.6 | 64.9 | 38.9 | 68.2 |
| Gemma | 7B | 41.6 | 60.3 | 42.0 | 70.3 | 45.7 | 67.1 | 39.2 | 63.4 | 40.0 | 70.6 | 35.6 | 56.3 | 48.0 | 69.3 | 42.6 | 70.2 |
| Average | N/A | 51.4 | 60.4 | 48.0 | 67.8 | 53.0 | 62.6 | 45.5 | 60.6 | 48.6 | 67.3 | 40.2 | 58.0 | 50.6 | 64.9 | 46.1 | 66.9 |

Table 12: Results of models in English by difficulty and group.

| Model | Variant | Easy | | Medium | | Hard | |
|---|---|---|---|---|---|---|---|
| | | Loc | Glo | Loc | Glo | Loc | Glo |
| GPT | 3.5 Turbo | 67.5 | 89.4 | 51.2 | 82.5 | 44.5 | 73.4 |
| | 4 | 78.3 | 94.4 | 67.8 | 91.7 | 62.1 | 87.4 |
| | 4 Turbo | 80.1 | 94.7 | 70.2 | 92.0 | 64.8 | 87.4 |
| Claude 3 | Haiku | 66.3 | 91.8 | 57.3 | 82.6 | 51.1 | 76.5 |
| | Sonnet | 66.3 | 91.8 | 54.4 | 85.7 | 53.4 | 80.6 |
| | Opus | 79.3 | 95.1 | 69.0 | 91.3 | 66.3 | 88.5 |
| Llama 2 | 7B | 44.9 | 72.4 | 41.0 | 63.5 | 38.0 | 55.2 |
| | 13B | 48.2 | 79.7 | 44.4 | 69.5 | 37.0 | 59.6 |
| | 70B | 57.5 | 86.2 | 47.6 | 77.7 | 40.7 | 66.8 |
| Latxa | 7B | 52.7 | 60.3 | 46.0 | 52.6 | 43.7 | 45.1 |
| | 13B | 63.8 | 75.5 | 55.8 | 65.9 | 48.7 | 59.3 |
| | 70B | 70.7 | 84.2 | 62.1 | 72.0 | 53.1 | 62.3 |
| Llama 3 | 7B | 46.8 | 67.8 | 43.4 | 64.1 | 36.2 | 55.8 |
| | 70B | 63.6 | 88.8 | 55.8 | 83.2 | 51.8 | 72.4 |
| Qwen 1.5 | 7B | 46.7 | 80.6 | 40.9 | 71.1 | 39.4 | 60.4 |
| | 14B | 51.6 | 85.0 | 43.1 | 76.8 | 38.0 | 63.2 |
| | 72B | 63.8 | 90.8 | 51.0 | 83.4 | 48.1 | 76.1 |
| Yi | 6B | 50.2 | 81.9 | 43.0 | 73.2 | 38.5 | 62.2 |
| | 9B | 51.3 | 83.9 | 42.1 | 74.8 | 36.8 | 64.0 |
| | 34B | 62.5 | 89.6 | 51.3 | 82.8 | 47.0 | 77.0 |
| Mistral | 7B | 55.3 | 82.3 | 46.1 | 74.4 | 39.5 | 63.5 |
| | 47B | 66.7 | 90.5 | 55.2 | 82.2 | 48.7 | 73.8 |
| Gemma | 7B | 53.2 | 84.7 | 43.0 | 76.9 | 39.8 | 65.3 |
| **Average** | N/A | 60.3 | 84.4 | 51.4 | 77.0 | 46.4 | 68.5 |

questions. For open models, results are more varied, but the relative drop is also generally bigger on local questions. On the most difficult local questions, some models get close to random chance.

# G   Extended Error Analysis

We include the remaining 25 examples of error analysis in Table 14.

Table 13: Results of models in English by difficulty and group.

| Model | Variant | Easy | | Medium | | Hard | |
|---|---|---|---|---|---|---|---|
| | | Loc | Glo | Loc | Glo | Loc | Glo |
| GPT | 3.5 Turbo | 54.7 | 69.9 | 45.6 | 66.5 | 40.0 | 61.3 |
| | 4 | 68.0 | 89.5 | 62.1 | 85.6 | 57.8 | 81.8 |
| | 4 Turbo | 78.0 | 91.9 | 66.8 | 89.1 | 62.1 | 86.0 |
| Claude 3 | Haiku | 67.0 | 87.4 | 56.0 | 80.7 | 50.0 | 69.2 |
| | Sonnet | 61.8 | 88.3 | 53.2 | 82.5 | 52.8 | 77.7 |
| | Opus | 79.2 | 93.5 | 68.8 | 91.2 | 64.7 | 87.1 |
| Llama 2 | 7B | 37.0 | 36.8 | 33.4 | 37.1 | 34.1 | 37.5 |
| | 13B | 32.4 | 46.8 | 36.1 | 42.9 | 33.7 | 41.1 |
| | 70B | 39.2 | 55.6 | 38.5 | 55.0 | 33.7 | 51.4 |
| Latxa | 7B | 56.5 | 55.0 | 48.3 | 51.6 | 42.1 | 44.4 |
| | 13B | 67.5 | 72.9 | 59.4 | 64.4 | 52.3 | 57.4 |
| | 70B | 75.8 | 82.3 | 64.6 | 70.8 | 51.5 | 61.3 |
| Llama 3 | 8B | 46.8 | 67.8 | 43.4 | 64.1 | 36.2 | 55.8 |
| | 70B | 63.6 | 88.8 | 55.8 | 83.2 | 51.8 | 72.4 |
| Qwen 1.5 | 7B | 37.3 | 47.4 | 35.4 | 46.3 | 35.0 | 44.4 |
| | 14B | 37.9 | 57.8 | 37.0 | 52.9 | 37.0 | 48.4 |
| | 72B | 46.5 | 64.0 | 42.7 | 65.5 | 38.2 | 59.5 |
| Yi | 6B | 38.5 | 49.0 | 39.1 | 46.8 | 35.8 | 42.8 |
| | 9B | 41.6 | 52.4 | 37.4 | 47.9 | 35.0 | 46.8 |
| | 34B | 46.4 | 59.4 | 40.6 | 60.8 | 34.9 | 61.3 |
| Mistral | 7B | 39.1 | 55.5 | 37.7 | 49.8 | 34.1 | 47.5 |
| | 47B | 48.0 | 64.4 | 40.6 | 62.0 | 41.9 | 55.6 |
| Gemma | 7B | 45.3 | 72.4 | 40.3 | 66.4 | 39.5 | 57.0 |
| **Average** | N/A | 52.5 | 67.3 | 47.1 | 63.6 | 43.2 | 58.6 |

Table 14: Last 25 error analysis examples annotated by difficulty of web search.

| Category | Diff. | Question | Candidate 0 | Candidate 1 | Candidate 2 | Ans. | Web |
|---|---|---|---|---|---|---|---|
| Sports and Leisure | 3 | Against whom did the "aizkolari" or woodchopper Joxe Mari Olasagasti first compete? | Against Floren Nazabal | Against Donato Larretxea | Against Jose Etxebeste | 2 | difficult |
| Music and Dance | 2 | Whose song is "Freaky Fiesta"? | The Skalariak group's | Betagarri's | Kortatu's | 1 | easy |
| Cinema and Shows | 2 | Who is the protagonist in Borja Cobeaga's short film "The First Time"? | Txema Blasco | Marivi Bilbao | Barbara Goenaga | 1 | difficult |
| Cinema and Shows | 1 | In the show "Vaya Semanita" where is the character El Jonan from? | From Otxarkoaga | From Zorroza | From Barakaldo | 2 | easy |
| Music and Dance | 3 | Where exactly is the Biscayan group "Izenik ez" from? | From Romo | Fron Abadiño | From Ugao-Miraballes | 0 | difficult |
| Cinema and Shows | 1 | Which festivity in the Basque Country appears in the film "Day & Night"? | The Aste Nagusia or Great Week of Bilbao | The Tamborrada of Donostia / San Sebastian | The San Fermín festival of Pamplona / Iruña | 2 | no |
| Music and Dance | 3 | Who currently directs the Alurr group in Ibarra? | Unax Sarriegi | Aiert Beobide | Edu Muruamendiaraz | 1 | no |
| Sports and Leisure | 2 | Where's the Onena leisure association from? | From Irun | From Ordizia | From Donostia / San Sebastian | 2 | difficult |
| Society and Traditions | 3 | Where is the oldest church in Biscay? | In Lekeitio | In Durango | In Bilbao | 1 | no |
| Music and Dance | 2 | Where's the Pi LT band from? | From the Mungia area | From Amorebieta-Etxano | From Bermeo | 0 | easy |
| Science and Technology | 2 | When was Pamplona / Iruña General Hospital founded? | In 1555 | In 1545 | In 1554 | 1 | no |
| Sports and Leisure | 2 | When is Navarre Day celebrated? | The first Sunday in May | The last Sunday in April | The last Saturday in April | 1 | difficult |
| Culture and Art | 1 | How are Sebastian and Joxe Lizaso related? | They are brothers | They are cousins | They are father and son | 2 | easy |
| Music and Dance | 2 | Where's the San Fermín dance company from? | From Zizur | From Tafalla | From Tudela | 0 | no |
| Cinema and Shows | 3 | Where was the Basque screenwriter and director Pello Varela born? | In Amorebieta-Etxano | In Vitoria-Gasteiz | In Portugalete | 1 | easy |
| Basque and Literature | 2 | How often is the Jakin magazine published? | Once a month | From time to time | Once a year | 0 | difficult |
| Basque and Literature | 3 | When did the translation and interpretation degree course start at the University of the Basque Country (UPV/EHU)? | In the 1995-1996 academic year | In the 1990-1991 academic year | In the 2000-2001 academic year | 2 | no |
| Science and Technology | 3 | What's the only amphibian on Mount Urgull? | The common frog | The salamander | The common toad | 1 | difficult |
| Geography and History | 1 | Where's the Gorbeialdea? | In Biscay | In Álava and Biscay | In Álava | 2 | easy |
| Music and Dance | 1 | Where's the Getaria dance company from? | From Labourd | From Álava | From Gipuzkoa | 0 | easy |
| Sports and Leisure | 3 | In 2010, which women's team won the Gipuzkoa trainera rowing championship? | San Juan | Getaria-Tolosa | Zumaia | 2 | no |
| Society and Traditions | 2 | Which of the following is not the function of the chartered councils? | Approval of the chartered regulations | Promotion of cultural activities | Tax collection | 0 | no |
| Basque and Literature | 3 | What was the first novel in Basque to be published in the Southern Basque Country after the War? | "Hiltzaileak" | "Alos-Torrea" | "Leturiaren egunkari ezkutua" | 1 | difficult |
| Sports and Leisure | 3 | Where is the Xaguxar leisure group from? | From Astigarraga | From Hernani | From Ordizia | 0 | easy |
| Music and Dance | 3 | Who was the first chairman of the Euskal Dantzarien Biltzarra (Association of Basque Dancers)? | Jesus Maria Arozamena | Jose Antonio Legarra | Juan Jose Garaizabal | 0 | difficult |

