## G  Dataset Information

**Dataset Card.**   The dataset card is included on this page `https://huggingface.co/datasets/HiTZ/BertaQA`.

**Croissant Metadata.**   The standard Croissant metadata of the dataset can be accessed at `https://huggingface.co/api/datasets/HiTZ/BertaQA/croissant`.

**Additional Metadata.**   To get dataset info: `https://datasets-server.huggingface.co/info?dataset=HiTZ/BertaQA` To get the number of rows and size in bytes: `https://datasets-server.huggingface.co/size?dataset=HiTZ/BertaQA`. To get statistics for each language check `https://datasets-server.huggingface.co/statistics?dataset=HiTZ/BertaQA&config=eu&split=test` for Basque and `https://datasets-server.huggingface.co/statistics?dataset=HiTZ/BertaQA&config=en&split=test` for English.

**Data hosting.**   We host the dataset on HuggingFace at `https://huggingface.co/datasets/HiTZ/BertaQA`. The data can be viewed in the HuggingFace dataset viewer. It can be loaded with the HF dataset loader. It can also be downloaded directly by cloning the HF repository or downloading individual files. It can be loaded directly as it is in the standard JSONL format.

**Code hosting.**   We host the code to reproduce experiments on GitHub: `https://github.com/juletx/BertaQA`. All instructions and code to reproduce our experiments are included in the repo.

**Licensing.**   We do not own any of the text from which this data has been extracted. We license the curation and translation of the dataset under CC-BY 4.0. The code is licensed under MIT license.

**Maintenance plan.**   We will ensure that the dataset and code are available for a long time. We are committed to maintaining the dataset and code to address any issues. We will actively monitor issues in the HuggingFace and Github repositories.

**Author statement.**   We bear all responsibility in case of violation of rights.