# OpenReview forum: "BertaQA: How Much Do Language Models Know About Local Culture?"
_NeurIPS.cc/2024/Datasets_and_Benchmarks_Track — NeurIPS 2024 Track Datasets and Benchmarks Poster_

### Official Review · Reviewer_7FUP · 2024-07-05

**Rating:** 6
**Confidence:** 4
**Correctness:** Yes
**Clarity:** Yes

**Review:**

- The datasets is based on "crawling public sources that are no longer available." Why are they no longer available? "Google does not return any result when searching for questions from the dataset verbatim." What efforts did the authors take to make this claim?

- No multilingual LLM is included since "XGLM and BLOOM" are bad (lines 99-100). Well, they are not so good at instruction following, but maybe stronger multilingual LLMs could be employed: for example Aya. Since this is specifically a paper about multilinguality and multiculturalism, maybe it would make sense to at least have one general multilingual LLM.

- How is the global subset designed? Since it's just general QA, why not employ existing QA datasets? What is unique in the design of this general subset?

- So the continued training on Basque is conducted by someone else and the authors evaluated the tuned models called Laxta. Maybe present more details about how their pretraining was done: what corpora was employed? Is it similar to the domains and topics in the BertaQA dataset? Even better, maybe the authors could conduct continued tuning of their own to go deeper into the cross-lingual transfer phenomenon. For example, analyze the intermediate checkpoints of continued pretraining and see when did the cross-lingual transfer happen: is it continuous or emergent?

- When I read the introduction I really liked the angle on local culture. However, there is very little "culture" in the analysis and experiments in the main paper. In addition to just reporting overall performance numbers over a bunch of models, maybe include some analysis/examples that focus on the culture aspect. For example, some culture-specific questions might be correctly answered in Basque but not in English; some might be correctly answered after continued tuning but not before; what's the proportion of these cases? Basically highlighting the cultural aspect of this dataset in addition to presenting a lot of numbers.

- Overall the scope and appeal of this work could be a bit narrow. Maybe it would be nice to highlight the general process/method to construct such as a dataset so that it's easily extensible to other languages/cultures, this would increase the impact of this work.

**Strengths:**

+ cultural evaluation is important
+ the dataset could be useful for Basque communities

**Additional Feedback:**

please see above

**Documentation:**

Yes

**Limitations:**

No limitations section in the main paper.

**Opportunities For Improvement:**

please see above

**Relation To Prior Work:**

Yes

**Summary And Contributions:**

This work presents BertaQA, a dataset evaluating the cultural knowledge of LLMs specifically focusing on the Basque culture. Experiments demonstrate consistent gaps between performance on Basque topics and general QA, while continued training with Basque corpora could help mitigate this gap.

---

> ### Author Rebuttal · Authors · 2024-08-19
>
> Thanks for the comments. We next address the main concerns raised in the review:
>
> > “The datasets is based on "crawling public sources that are no longer available." Why are they no longer available? "Google does not return any result when searching for questions from the dataset verbatim." What efforts did the authors take to make this claim?”
>
> The Basque portion of the dataset was created by compiling trivia questions from the public web, but the corresponding website is no longer available for reasons that are unknown to us. To check whether the content is present in other websites, we wrote some of the questions verbatim in Google search using quotation marks, but received no results. We will make these points more clear in the paper.
>
> > “No multilingual LLM is included since "XGLM and BLOOM" are bad (lines 99-100). Well, they are not so good at instruction following, but maybe stronger multilingual LLMs could be employed: for example Aya. Since this is specifically a paper about multilinguality and multiculturalism, maybe it would make sense to at least have one general multilingual LLM.”
>
> All the open models we include are base models, not instruct. They just need to have in-context learning abilities to be able to answer the questions. However, we agree that including a stronger multilingual LLM such as Aya makes sense, despite being an instruction-tuned model. We could also extend the evaluation to include more open instruction models, but we found no significant differences in some preliminary experiments. Next we show the results for Aya-101 13B.
> ```
> Basque
> Global: 55.60
> Local: 42.39
> English
> Global: 57.57
> Local: 41.71
> ```
> We see that it on par with Llama 3 8B only in Basque on Local questions, while being much worse in all other settings. Despite not being very multilingual, Llama 3 is not bad in Basque, the 70B model results are very close to the English results, there is only a two point difference.
>
> > “How is the global subset designed? Since it's just general QA, why not employ existing QA datasets? What is unique in the design of this general subset?”
>
> As mentioned above, the Basque portion of the dataset was created by compiling trivia questions from the public web. The questions that we compiled were already divided between global and local topics. We did inspect the dataset and confirmed that the division was well-founded. The motivation for using this global subset, as opposed to existing QA datasets, is that the questions come from the same source, so the results should be more comparable.
>
>  > “So the continued training on Basque is conducted by someone else and the authors evaluated the tuned models called Laxta...”
>
> The Latxa model was developed in our research group. In fact, three authors of this paper are also authors of it, and the first author is the same. However, we consider that to be a separate work, which is why we did not discuss it in depth and instead refer the reader to Etxaniz et al. (2024). Having said that, we agree that describing how Latxa was trained in more detail would be helpful to interpret our results. Addressing the reviewer’s question, Basque is a low-resource language, so Latxa was trained on all publicly available corpora in Basque meeting some minimum quality standards. This mostly corresponds to crawling corpora, including both processed versions of CommonCrawl as well as ad-hoc crawling of websites with high-quality content. The largest portion of it consists of news, which is the most common use of Basque on the web. While the resulting pretraining corpus is diverse in nature, we would not say that the domains and topics in BertaQA are particularly prominent on it, although it is obviously more likely that topics related to the Basque culture are discussed in the Basque language compared to English. We will include this additional context in the paper.
>
> > “When I read the introduction I really liked the angle on local culture. However, there is very little "culture" in the analysis and experiments in the main paper....”
>
> We agree with the reviewer that more analysis and examples of the behavior of the models, including an error analysis of the mistakes that models do in cultural-specific dataset. We did a preliminary analysis of the most difficult questions of the culture-specific dataset. Look at the answer to reviewer NEZ825 for some examples of those questions that GPT4 Turbo answers incorrectly. Next we include some results by category where we can see that the model is generally better in Basque in 3 categories: Society and Habits, Music and Dance and Science and Technology. We will extend this preliminary analysis to answer your questions.
>
> | Language | Basque and Literature | Geography and History | Society and Habits | Sports and Leisure | Culture and Art | Music and Dance | Science and Technology | Cinema and Shows |
> |----------|-----------------------|-----------------------|--------------------|--------------------|-----------------|-----------------|------------------------|------------------|
> | EU       | 67.0                  | 61.3                  | 73.1               | 51.3               | 70.0            | 43.8            | 63.3                   | 65.8             |
> | EN       | 73.2                  | 66.3                  | 70.5               | 58.8               | 80.0            | 36.3            | 58.2                   | 72.2             |
>
> > “Overall the scope and appeal of this work could be a bit narrow. Maybe it would be nice to highlight the general process/method to construct such as a dataset so that it's easily extensible to other languages/cultures, this would increase the impact of this work.”
>
> We will include additional details about how the dataset was constructed. Please refer to the answer to reviewer NEZ825 and SVZZ23 for more details.

---

> > ### Comment · Reviewer_7FUP · 2024-08-20
> >
> > I would like to thank the authors for their response. I have adjusted the rating.

---

### Official Review · Reviewer_SVZZ · 2024-07-23
**More detailed documentation required and limited scope**

**Rating:** 6
**Confidence:** 4

**Review:**

The paper provides significant insights into the interplay between language, culture, and knowledge in LLMs. The originality lies in the focus on minority cultures. The paper is clear and well-organized. The introduced dataset is of high quality. However, as the authors mentioned, the scope is somewhat limited to Basque culture. More importantly, a more detailed documentation on data collection, categorization and translator quality would enhance the study's reproducibility and validity. Nevertheless, the novel insights of this paper outweigh its limitations, warranting its acceptance.

**Strengths:**

1. The dataset includes professionally translated samples, ensuring high-quality parallel data in English and Basque.
2. The inclusion of both local and global subsets allows for a nuanced comparison of LLM performance in context of low-resource culture-specific knowledge capabilities.
3. Additionally, the evaluation of models in both English and Basque along with cross-lingual learning techniques provides a comprehensive view of cross-lingual capabilities and limitations and further insights of knowledge transfer from low-resource language data.

**Additional Feedback:**

- The authors should provide evidence for the statement in line 139 regarding the correlation between local and global performance metrics.
- More details about the data source, including when it was crawled and the nature of the original sources, would enhance the transparency of the dataset creation process.
- It would be valuable to discuss how the findings can be transferred to other cultures and societies beyond the Basque culture, offering insights for broader applications.

**Clarity:**

The paper is well-written, with clear explanations.

Minor Comment:
A missing space in the Table 3 caption ("ofBERTAQA").

**Correctness:**

The claims made in the submission are generally correct, supported by extensive experimental results. However, the support for some statements, such as the above mentioned strong correlation (4 in Opportunities For Improvement), could be improved with additional evidence.

**Documentation:**

The documentation is generally good but could be improved by providing more details about the data source, categorization process, and translation quality assurance.

**Ethics:**

There are no significant ethical concerns with this study.

**Limitations:**

The authors adequately address the limitations in the conclusion and future work section.

**Opportunities For Improvement:**

1. The study's scope is limited to Basque culture, reducing its generalizability to other (minority) cultures. As the authors mention, future work should aim to include data/knowledge from multiple cultures to validate the findings across diverse contexts.
2. The paper provides a sparse description of the data categorization and difficulty labeling process. More detailed explanations and justifications for these categorizations would strengthen the benchmark.
3. The quality assurance process for translations is not thoroughly documented. For instance, including more information about the translators and how translation quality was ensured would add credibility.
4. The statement “the training recipes of existing models are roughly equivalent in how they balance global and local knowledge“ (line 141) lacks robust support. Providing more detailed evidence or metrics for the strong correlation would substantiate this claim.

**Relation To Prior Work:**

The paper clearly differentiates itself from previous work. The discussion of prior work is thorough and well-contextualized.

**Summary And Contributions:**

This paper introduces BERTAQA, a multiple-choice benchmark designed to evaluate the performance of LLMs on knowledge acquisition from minority cultures, specifically in this case Basque. The introduced dataset includes samples in both English and Basque, including local questions (covering basque specific knowledge) and questions on globally known knowledge, in total covering 8 categories with varying levels of difficulty. The study finds that, not surprisingly, LLMs perform better on global topics than on local ones. Next evaluating the LLMs on the cultures corresponding language the paper also evaluates the LLMs on the translation of the dataset to English. Further, the paper explores cross-lingual learning techniques, demonstrating novel insights of the potential for knowledge transfer from low-resource languages to high-resource ones and highlighting the limitations of current translation-based approaches.

---

> ### Author Rebuttal · Authors · 2024-08-19
>
> Thanks for the valuable feedback. We are glad that the reviewer believes that “the novel insights of this paper outweigh its limitations, warranting its acceptance”. We next address the concerns raised in the review:
>
> > “The study's scope is limited to Basque culture, reducing its generalizability to other (minority) cultures.”
>
> We agree with the reviewer, and acknowledge this limitation in the paper itself (lines 272-276). Having said that, we believe that knowing the corresponding language and culture is critical to ensure the quality of this type of datasets, which made it unfeasible for us to create similar datasets for other minority cultures. In addition, we believe that the finding that LLMs can behave differently in global and local topics is relevant besides the specific case of Basque. It is true that we only show this difference for the Basque culture, but the finding that such a difference can exist prompts to reconsider how LLMs are evaluated more broadly, and motivates the creation of similar datasets for other minority cultures.
>
> > “The paper provides a sparse description of the data categorization and difficulty labeling process.” and “The quality assurance process for translations is not thoroughly documented.”
>
> We will extend Section 2 to include more details about the creation process of the dataset. As discussed above, the Basque portion of the dataset was created by compiling trivia questions from the public web. The questions were already classified into local and global topics, and labeled according to their difficulty level and knowledge category. As such, no human annotator was involved in the creation of the Basque portion of the dataset. In order to create the English portion, we translated the Basque data through a professional translation company (Elhuyar itzulpenak, https://itzulpenak.elhuyar.eus/en). We first wrote some translation guidelines, covering things like formatting or how to translate Basque named entities. In addition, our guidelines asked translators to discard questions whose answers require knowing the Basque language (such as onomatopeias). We initially sent 100 question/answers for translation. We reviewed these translations carefully, and worked closely with the professional translators to clarify and extend the guidelines accordingly. The remaining dataset was translated in batches of 1000 samples. The translators tagged problematic samples (difficult translation, outdated information, more than one correct answer…), which we manually reviewed. We will include these details in the paper.
>
> > “The statement “the training recipes of existing models are roughly equivalent in how they balance global and local knowledge“ (line 141) lacks robust support.”
>
> As discussed above, it was not our intention to formulate this as a claim, but rather as a plausible explanation of the pattern observed in Table 2 (which is why we phrase it as “this suggests” as opposed to “this proves”). More concretely, Table 2 shows that models with similar scores in local topics tend to obtain similar scores in global topics (the Pearson correlation between the two scores is 0.844, which is very high). We presume that, if the training corpus of a given model was significantly more skewed towards local topics than that of another model, the former would tend to perform better in local topics, at least in relative terms. Given that we do not observe this, we hypothesize that the training recipes of existing models are roughly equivalent in how they balance global and local knowledge. In any case, we understand that our statement could look unjustified, so we will soften it and include this additional context.
>
> > “Minor Comment: A missing space in the Table 3 caption ("ofBERTAQA").”
>
> Thanks, we will correct it.

---

> > ### Comment · Reviewer_SVZZ · 2024-08-28
> >
> > Thank you for the response, especially for clarifying and providing additional information on the labeling and translation process. I don't have further questions.

---

### Official Review · Reviewer_NEZ8 · 2024-07-25
**Relevant dataset, results are not very surprising**

**Rating:** 6
**Confidence:** 4

**Review:**

BERTAQA is a relevant dataset for measuring the performance of LLMs for cultural specific questions in Basque - the topic of creating datasets specific to different cultures in the world is an important topic. I like that the dataset is quite large, on the other hand the fact that is just a multi-choice QA (MCQA) dataset makes it less useful for the community as many cultural aspects are not factoid and therefore cannot be measured by MCQA alone.

The dataset is large for the task at hand, it has questions in various categories and three difficulty levels, and the authors claim that the answers are not (easily?) found using online search. On my side, the important part are the local questions related to the Basque culture, as for global cultural questions I feel this is a more difficult task and the dataset should be larger. But the global data can be used as a starting point and as a reference to compare the difference in performance compared to local questions. Overall, the results indicate an important decrease in performance for local questions, in English or in Basque, with results being poorer in Basque as expected.

The main weaknesses of the paper are twofold. First, for a dataset paper, there is so little information on human annotators and the entire dataset creation process. Second, the results section is pretty long and while it makes sense, I have hardly been able to find any surprising results (meaning I would have expected somehow the reported performance without even reading the paper). I would have liked a more thorough analysis of the most difficult question - I think that would make the paper more relevant and provide original results.

**Strengths:**

* Very relevant dataset for measuring the performance of LLMs on cultural specific questions
* The comparative results section includes a wide range of LLMs
* Dataset is available both in Basque and professionally translated to English, contains different categories and levels of difficulty

**Additional Feedback:**

No other feedback.

**Clarity:**

The paper is clear and well-structured. There are a couple of mistypes, e.g. "English version ofBERTAQA"

**Correctness:**

In general, the paper is sound. However, I cannot agree with the following statement without much more evidence:
- How did the authors check that the questions in BERTAQA cannot be found using online search? This is a very strong statement and I assume that after reading several online documents found using search, one would find the answer in the end.

**Documentation:**

The Github page is good.

**Ethics:**

No ethical concerns.

**Limitations:**

Most of the limitations are covered, but I would have liked to see some mention of bias especially as there are so few details about annotators.

**Opportunities For Improvement:**

I think the paper has some important directions of improvement I expect the authors to try and implement them for the revised / final version.

- Most important: the dataset annotation process needs to be discussed in more details: who are the human annotators, were there any quality filters, how much did you pay the annotators, how did you check for annotator bias in creating questions, were annotators paid?
- I would have liked a more thorough analysis of the most difficult question - I think that would make the paper more relevant and provide original results.
- The authors mention that Latxa mainly has pretraining improvements for Basque; does it also have any alignment improvements?

**Relation To Prior Work:**

Yes, prior work is well discussed. There is a recent cultural-specific dataset proposed to assess the quality of LLMs on Romanian. The authors propose a dataset similar to MT-Bench [1], not MCQA style. Maybe it is useful to update the final version of the prior work section and show that other languages started building similar cultural-probing datasets.

[1] - https://arxiv.org/abs/2406.18266

**Summary And Contributions:**

The paper presents a new question answering dataset, called BERTAQA, which aims to measure the performance of LLMs on questions specific to a culture, in this case Basque. I consider that it is really important to have datasets specific to different cultures in the world, in order to be able to understand not only the performance on this cultural-specific tasks, but also if there are biases and to investigate the weak spots of the models - maybe to also understand why this happens.

While the dataset and task is relevant, the paper does not go into many details and just provides a very comprehensive comparison of the results for various models, most of them general and multilingual models and a Llama-2 specific model pretrained on a Basque corpora. Most (all) of the results are not very surprising (including the "solid evidence of knowledge transfer from a low resource to a high-resource language" that the authors are claiming) as it is expected to have transfers in any directions and if the information pertinent to the questions in BERTAQA is in Basque, there will be an improvement in English as well (or in other languages).

I would have liked a more thorough analysis on the mistakes the models make, e.g. which questions are difficult for the top models. This would have provided more value to understanding how to create difficult questions for cultural-specific datasets. At the same time, I feel another important week point for a dataset specific paper is the lack of information about the human annotators and the entire dataset creation process.

---

> ### Author Rebuttal · Authors · 2024-08-19
>
> Thanks for the detailed review. Please find our answers below:
>
> > “Most important: the dataset annotation process needs to be discussed in more details: who are the human annotators, were there any quality filters, how much did you pay the annotators, how did you check for annotator bias in creating questions, were annotators paid?”
>
> The Basque portion of the dataset was created by compiling trivia questions from the public web. The questions were already classified into local and global topics, and labeled according to their difficulty level and knowledge category. As such, no human annotator was involved in the creation of the Basque portion of the dataset. In order to create the English portion, we translated the Basque data through a professional translation company (Elhuyar itzulpenak, https://itzulpenak.elhuyar.eus/en). We first wrote some translation guidelines, covering things like formatting or how to translate Basque named entities. In addition, our guidelines asked translators to discard questions whose answers require knowing the Basque language (such as onomatopeias). We initially sent 100 question/answers for translation. We reviewed these translations carefully, and worked closely with the professional translators to clarify and extend the guidelines accordingly. The remaining dataset was translated in batches of 1000 samples. The translators tagged problematic samples (difficult translation, outdated information, more than one correct answer…), which we manually reviewed. We will include these details in the paper.
>
> > “I would have liked a more thorough analysis of the most difficult question - I think that would make the paper more relevant and provide original results.”
>
> We manually inspected a random subset of hard questions that GPT 4 Turbo answered incorrectly. We were unable to find any clear pattern, besides most questions being very specific and somewhat niche. We next include 10 examples of the hard questions. If the paper is accepted, we will extend this preliminary analysis and include it in the camera ready version.
>
> ```
> Which of these towns does not have a common boundary with Elorrio?
> a) Zaldibar
> b) Bergara
> c) Matiena
>
> Which city in the Continental Basque Country is Pamplona / Iruña twinned with?
> a) Donibane Garazi - Saint-Jean-Pied-de-Port
> b) Bayonne
> c) Mauleon
>
> Which of these districts is not in the Pettarra area of Soule?
> a) Etxarri
> b) Garrüze
> c) Sarricotapea
>
> In spring and summer, what percentage of extra hours of sunlight do they have in La Rioja compared with Bilbao?
> a) 33%
> b) 22%
> c) 44%
>
> Who designed the Bilbao Museum of Fine Arts?
> a) Fernando Urrutia
> b) Migel Urrutia
> c) Peru Urrutia
>
> After the release of four records, what year did the band "Baldin Bada" of Irun break up?
> a) In 1996
> b) In 2007
> c) In 2003
>
> What year was "La Vizcaya" factory set up?
> a) In 1882
> b) In 1884
> c) In 1886
>
> Which king of Navarre died in 882?
> a) Fortun Gartzia
> b) Eneko Aritza
> c) Gartzia Eneko
>
> When was the Leurtza reservoir built?
> a) In 1920
> b) In 1925
> c) In 1921
>
> In what year did King Liuvigild found the city of Vitoria-Gasteiz?
> a) In 578
> b) In 581
> c) In 576
> ```
>
> > “The authors mention that Latxa mainly has pretraining improvements for Basque; does it also have any alignment improvements?”
>
> Latxa is a base model, and it has not been aligned or post-trained to follow human instructions. We will make this point more clear in the paper.
>
> > “There is a recent cultural-specific dataset proposed to assess the quality of LLMs on Romanian. The authors propose a dataset similar to MT-Bench [1], not MCQA style.”
>
> Thanks for the pointer. We were not aware of this work, but we agree that it is relevant and we will discuss it in the related work section.

---

> > ### Comment · Reviewer_NEZ8 · 2024-08-19
> >
> > Thank you for the answers. Given that the questions are from trivia questions from the public web, how can you comment related to the following point?
> >
> > - How did the authors check that the questions in BERTAQA cannot be found using online search? This is a very strong statement and I assume that after reading several online documents found using search, one would find the answer in the end.
> >
> > I am sure the final version of the paper would benefit from an analysis of the difficult questions, even the initial insights from the above comment seem useful.

---

> > > ### Author Rebuttal · Authors · 2024-08-20
> > >
> > > The Basque portion of the dataset was created by compiling trivia questions from the public web, but the corresponding website is no longer available for reasons that are unknown to us. To check whether the content is present in other websites, we wrote some of the questions verbatim in Google search using quotation marks, but received no results. Finding the answer to these questions on the web is possible, but at least the same questions are not present on the web. We will make these points more clear in the paper.

---

### Official Review · Reviewer_Zd5e · 2024-08-03
**Important dataset on a low resource language**

**Rating:** 7
**Confidence:** 4
**Clarity:** Well written.

**Review:**

An interesting study which shows the weaknesses of LLMs regarding local culture and show that they are mainly trained on global English culture.

**Strengths:**

- Multilingual, English and Basque dataset
- Interesting study showing performance on global and local culture

**Additional Feedback:**

Generally, it is a good paper, only a revision with rephrasing the claims and it would be ready to publish.

**Correctness:**

It is constructed correctly and the comments about the claims are mentioned above.
The design is appropriate with no problems.

**Documentation:**

No problem with the documentation

**Ethics:**

No problem with Ethics

**Limitations:**

The claims and conclusions need to be revised as they are too general with the scheme of the paper
The paper is only evaluating Basque which is important and needed, but claims, based on it, can not generalize to other different languages to the extend of writing such conclusions.

**Opportunities For Improvement:**

the claims and conclusions are too general without more inspection, this need to be revised and corrected accordingly.
Line 141 in page 4 is a claim with no real justification.
Line 144 in page 5, why describing LLama as weaker, how do the authors claim weaker vs stronger models.
Line 173, training on other language may not be really harming English, this claim need more than one language and more than the claimed model to be able to suggest this claim.

**Relation To Prior Work:**

enough mentioned related work

**Summary And Contributions:**

The paper presents a dataset that evaluate the performance of LLMs on local and global information through MCQ  and show their performance in English and Basque as well.

---

> ### Author Rebuttal · Authors · 2024-08-19
>
> Thanks for the feedback. We next address the main concerns raised in the review:
>
> > “Line 141 in page 4 is a claim with no real justification.”
>
> It was not our intention to formulate this as a claim, but rather as a plausible explanation of the pattern observed in Table 2 (which is why we phrase it as “this suggests” as opposed to “this proves”). More concretely, Table 2 shows that models with similar scores in local topics tend to obtain similar scores in global topics (the Pearson correlation between the two scores is 0.844, which is very high). We presume that, if the training corpus of a given model was significantly more skewed towards local topics than that of another model, the former would tend to perform better in local topics, at least in relative terms. Given that we do not observe this, we hypothesize that the training recipes of existing models are roughly equivalent in how they balance global and local knowledge. In any case, we understand that our statement could look unjustified, so we will soften it and include this additional context.
>
> > “Line 144 in page 5, why describing LLama as weaker, how do the authors claim weaker vs stronger models.”
>
> Thanks for pointing this out, we agree that this statement requires some justification. To clarify, we refer to Llama 2 as weaker because it is the model family obtaining the worst results in Table 2 relative to its size (e.g., Llama 2 7B is worse than all other 6-8B models in both local and global topics). To avoid any confusion, we will rewrite the sentences as follows: “For model families with the lowest scores in BertaQA, like Llama 2, scaling yields bigger gains on the global subset, as the delta between local and global questions increases from 22.80 for the smallest variant to 28.53 for the largest variant. The opposite is true for more performant model families like GPT, with the delta between local and global questions going from 27.32 for GPT-3.5 Turbo to 19.51 for GPT-4 Turbo.”
>
> > “Line 173, training on other language may not be really harming English, this claim need more than one language and more than the claimed model to be able to suggest this claim.”
>
> While we think that the reviewer raises a valid concern, we also feel that it is misquoting our work. We do not literally state that “training on other language may not be really harming English”. Instead, the paper says that “prior work mostly evaluated on global topics”, which “led to the incomplete conclusion that training on other languages harms English” (lines 172-174). We feel that the first part of this statement (“prior work mostly evaluated on global topics”) is well supported (see related work), and the reviewer does not seem to question it. Regarding the second part, our work does demonstrate that training on Basque does not harm English when evaluated on topics related to the Basque culture. It is true that we do not prove this for other languages and models. But we think that it is fair to say that the “conclusion that training on other languages harms English” is incomplete for these other languages and models until they are evaluated on local topics, as we find compelling evidence that the behavior on local topics can be fundamentally different for at least one language and model. In other words, our work shows that evaluating on global topics alone does not always show the full picture, which is what we intended to claim here (although the reviewer is right that evaluating on local topics might not necessarily show a different picture for other languages and models). In any case, we understand that our current phrasing can be misleading, so we plan to rewrite it as follows: “Given that prior work mostly evaluated on global subjects, this led to the generally accepted conclusion that training on other languages harms English. We show that this conclusion does not necessarily show the full picture, since models have barely been evaluated on local topics, and their behavior there can be fundamentally different.”

---

### Comment · Area_Chair_5WoF · 2024-08-29
**Please engage in the author-reviewer discussion**

Dear Reviewers,

Thank you for your hard work on the papers and reviews. Please note that the deadline for the author-reviewer discussion period is approaching (August 31, 2024). Some of you have engaged in discussions with authors-thank you! For reviewers who have not yet, please discuss with authors as this is a very important part of the reviewing process and authors are eager to have further feedback from you. If there are any changes to your scores, kindly provide explanations for these adjustments.

---

### Decision · Program_Chairs · 2024-09-26

**Decision:**

Accept (Poster)

**Comment:**

The paper presents a new QA dataset, BERTAQA, aiming to measure the performance of LLMs on questions specific to local culture.

Pros:
1. Evaluating LLMs with respect to cultures is important and interesting.
2. The dataset is available in both Basque and English, covering different categories and levels of difficulty.
3. The paper evaluates a wide range of LLMs.

Cons:
1.  The details on human annotations are missing.
2. It would be better to cover more languages and cultures.
3. More in-depth analysis and insights could be provided in the paper.